# Genome-Wide Characterization and Functional Analysis of *CsDOF* Transcription Factors in *Camellia sinensis* cv. Tieguanyin Under Combined Heat–Drought Stress

**DOI:** 10.3390/plants14121829

**Published:** 2025-06-14

**Authors:** Yingxin Wen, Cunyi Tan, Yujie Zhang, Hua Wu, Dian Chen, Heng Yue, Zekai Ding, Shijiang Cao, Kehui Zheng

**Affiliations:** 1College of Forestry, Fujian Agriculture and Forestry University, Fuzhou 350002, China; 18050409132@139.com (Y.W.); 18965989766@139.com (D.C.); 13731447144@139.com (Z.D.); 2College of Life Science, Fujian Agriculture and Forestry University, Fuzhou 350002, China; tancunyi133@gmail.com (C.T.); 19154658830@163.com (H.Y.); 3College of Horticulture, Fujian Agriculture and Forestry University, Fuzhou 350002, China; yu1444z@163.com; 4College of Food Science, Fujian Agriculture and Forestry University, Fuzhou 350002, China; w2918083657@163.com; 5College of Computer and Information Sciences, Fujian Agriculture and Forestry University, Fuzhou 350002, China

**Keywords:** tea, Tieguanyin, DOF, gene family, abiotic stress, expression analysis

## Abstract

Tieguanyin tea, celebrated as one of China’s top ten famous teas, is highly regarded for its unique flavor and taste. However, recent intensification of global warming has escalated the occurrence of abiotic stresses, posing significant threats to the growth, development, yield, and quality of Tieguanyin tea plants. DOF (DNA-binding one zinc finger protein), a plant-specific transcription factor, plays a critical role in plant development and stress response. In this study, we identified and analyzed 58 *CsDOF* genes across the whole genome, which were found to be randomly and unevenly distributed across 15 chromosomes. A phylogenetic tree was constructed using *DOF* genes from *Arabidopsis thaliana* and Tieguanyin, categorizing these genes into 10 subgroups. Collinearity analysis revealed homologous gene pairs between *CsDOF* and *OsDOF*(19 pairs), *StDOF* (101 pairs), and *ZmDOF* (24 pairs). Cis-acting element analysis indicated that *CsDOF* genes contain elements related to both stress and hormone responses. Heat map analysis demonstrated that subfamily C2 predominantly regulates the growth and development of roots, stems, and leaves in Tieguanyin. Tertiary structure analysis of CsDOF proteins revealed diverse structures, underscoring the functional variability within the *CsDOF* gene family. Furthermore, qRT-PCR analysis was employed to assess the expression profiles of 13 *CsDOF* genes under high-temperature and drought conditions. Notably, *CsDOF51* and *CsDOF12* exhibited significant expression changes under drought and high-temperature stress, respectively, while *CsDOF44* showed significant changes under both conditions. This study provides foundational knowledge of the *CsDOF* gene family and offers novel insights for enhancing the drought and heat tolerance of Tieguanyin tea.

## 1. Introduction

In recent years, with the continuous deterioration of the environment, plants have faced more severe survival problems. A series of abiotic stresses significantly affect the growth and development of plants. In this context, plants have evolved corresponding morphological structures and mechanisms to protect themselves [1,2,3]. Transcription factors, also known as trans-acting factors, are a group of protein molecules that specifically bind to particular DNA sequences, thereby ensuring the precise expression intensity of target genes in specific temporal and spatial contexts. They regulate the expression of the majority of genes and are involved in diverse biological functions [4,5]. These transcription factors not only enhance plant resistance to abiotic stresses, including heat stress [6], cold stress [7], drought [8], salinity [9], and heavy metals [10,11], but also facilitate plant growth, development [12,13], and fruit ripening [14]. Therefore, the study of transcription factors plays a key role in helping plants cope with abiotic stress and enhancing plant stress resistance.

The DNA-binding one zinc finger (DOF) protein is a plant-specific transcription factor belonging to the C2C2-type zinc finger family [15]. DOF proteins, generally composed of 200–400 amino acids, possess a highly conserved DOF domain comprising 50–52 amino acids [15]. Notably, four cysteine residues are located within the conserved region of this domain, forming a zinc finger structure critical for DNA binding [15]. The core recognition sequence of DOF proteins is the AAAG motif, and flanking sequences can influence binding affinity [16]. Additionally, the DOF domain exhibits bifunctional activity, enabling both DNA binding and protein–protein interactions [15]. The N-terminal region contains the conserved DOF domain [17], while the C-terminal region is variable and regulates transcription by interacting with regulatory proteins or signaling factors, contributing to functional diversity among DOF transcription factors [18,19]. DOF plays important roles in seed and seedling development [20], tissue differentiation [21], organ development [22], fruit ripening [23], and abiotic stress responses [24]. Furthermore, DOF transcription factors participate in carbon and nitrogen metabolism [17], regulate light signaling [25], mediate hormone responses [26], influence biomass synthesis and accumulation [27,28], and affect male fertility [29]. In addition, Tieguanyin is a self-incompatible species characterized by high heterozygosity, which poses significant challenges for genome assembly. Previous studies estimated its genome size to be approximately 2.98–3.1 Gb, with N50 contig lengths ranging from 67 kb to over 1 Mb. These genomic features highlight the complexity of studying the molecular mechanisms of tea plants under non-biological stress conditions [30].

Tieguanyin, a type of oolong tea, is a traditional Chinese tea, originally produced in Anxi County, Fujian Province. Because of its unique taste and health benefits, Tieguanyin has a broad market and significant economic value in China and even the world [31,32]. It is also an important economic crop. Its health benefits are mainly reflected in the prevention and relief of Alzheimer’s disease that result from drinking Tieguanyin tea [33]. Additionally, phenolic-rich compounds extracted from Tieguanyin exhibit significant antioxidant activity, demonstrating potential applications in the food and pharmaceutical industries [34]. However, in recent years, global warming has increased the frequency and intensity of non-biological stresses such as drought and high temperatures, posing a severe threat to the growth, development, and quality of tea plants [35,36]. Under drought stress, the root water absorption capacity of tea plants decreases [37], and changes in the leaf cuticle affect transpiration [38], leading to cell dehydration [39], inhibition of photosynthesis, and oxidative damage [40]. Ultimately, this results in inhibited growth of tea plants and impaired synthesis of internal substances in tea leaves [41,42,43]. Therefore, it is necessary to explore the resistance gene resources of Tieguanyin and analyze its molecular regulatory network in response to high temperatures and drought. Predecessors have demonstrated that DOF transcription factors play significant roles in plant stress responses. For instance, *ZmDOF22* in maize enhances drought tolerance by promoting stomatal closure to reduce water loss under drought stress [44]. Similarly, analysis of kiwifruit *DOF* genes revealed that six *AcDOFs* are differentially involved in abiotic stress responses, with *AcDOF22* overexpression reducing water loss and delaying chlorophyll degradation, thereby improving drought tolerance [45]. Under cold stress, 12 out of 25 *VaDOF* genes in grapevines were upregulated, among which *VaDOF17d* was identified as a key regulator of raffinose family oligosaccharides (RFOs) to confer cold resistance [46]. In orchids, class I genes (*CgDOF02, CgDOF07, CgDOF17*, and *CgDOF23*) exhibited potential for heat stress tolerance [47], while most *PeDOF* members in passion fruit were upregulated under heat stress, except *PeDOF4/5/10/13*, with *PeDOF11* showing the highest expression [48]. Under cadmium stress, *RsDOF33* in radish acted as a negative regulator [49]. In lotus, *NcDOF* genes displayed tissue-specific expression patterns, with most members being highly sensitive to salt stress [50]. Studies have shown that the *DOF* gene is involved in the abiotic stress response of Tieguanyin. However, the exact mechanism behind this process is still poorly understood. This study identified 58 *CsDOF* genes in Tieguanyin and analyzed their physicochemical properties, gene structures, chromosomal distribution, promoter cis-elements, and evolutionary relationships. Based on the results of qRT-PCR, we also detected the expression of *CsDOF* genes in different tissues and analyzed the differences in the expression levels of *CsDOF* genes under drought and high-temperature stress. These findings lay a foundation for further study of the *DOF* gene in Tieguanyin.

## 2. Results

### 2.1. Comprehensive Analysis of Chromosomal Distribution and Protein Physicochemical Properties of CsDOF Gene Family in Tieguanyin

Chromosome mapping showed that 58 *CsDOF* genes were widely and unevenly distributed on 15 chromosomes (Figure 1 and Table 1). Chromosomes including Chr04HA, Chr08HA, and Chr08HB exhibited distinct gene clustering, a higher density, and selective enrichment of *CsDOF* genes. However, regions including Chr05HA, Chr05HB, and Chr07HB showed sparse or absent *CsDOF* gene distribution, indicating regional depletion.

The observed gene clustering in enriched regions might reflect transcriptionally active chromatin domains, whereas the sparse distribution in other regions could correspond to evolutionarily conserved or functionally specialized genomic segments. To characterize these genes products further, we analyzed their protein physicochemical properties comprehensively. Table 1 summarizes the molecular characteristics of CsDOF proteins, including protein ID, polypeptide length (aa), molecular weight (Da), theoretical pI, instability index (II), aliphatic index (AI), and GRAVY value. The CsDOF protein family displayed significant variation in polypeptide length (161–479 aa) and molecular weight (18,141.50–52,129.17 Da). Theoretical isoelectric point (pT) analysis showed a broad range (4.77–9.61), with most CsDOF proteins being basic (pI > 7.0). Instability index (II) values ranged from 39.9 to 73.7, with only one member below the threshold of 40. This distribution pattern indicates that most CsDOF proteins exhibit inherent structural instability and are potentially prone to proteolytic degradation. Additionally, the aliphatic index ranged from 32.39 to 79.87, while the grand average of hydropathicity (GRAVY) values spanned from −1.204 to −0.338, with all CsDOF members displaying negative GRAVY values, consistent with the overall hydrophilic nature of the gene family. Comparative analysis shows that many CsDOF proteins have the same physical and chemical properties, including polypeptide length, molecular weight, isoelectric point, instability index, aliphatic index, and GRAVY value, strongly indicating structural and functional similarity with potential evolutionary significance.

### 2.2. Phylogenetic Analysis and Functional Diversification of DOF Gene Family in Arabidopsis and Camellia sinensis (CsDOF)

This study utilized MEGA 7.0 software to construct a phylogenetic tree of the *DOF* gene family in *Arabidopsis thaliana* and tea (*CsDOF*) using the maximum likelihood method (Figure 2). Based on the phylogenetic tree and previous research findings, we assigned consecutive numbers to genes within the same branch, from A to D. The genome sequences were divided into nine distinct subgroups (A, B1, B2, C1, C2.1, C2.2, D1.1, D1.2, D2, and C3). Key findings are summarized as follows:

A total of 58 candidate *CsDOF* genes were identified in *Camellia sinensis* cv. Tieguanyin, which were systematically designated as *CsDOF1* to *CsDOF58* sequentially. To further analyze these screened sequences, a phylogenetic tree comprising these 58 sequences was constructed. Based on the phylogenetic tree, the relevant genomic sequences were classified into nine subfamilies: A, B1, B2, C1, C2.1, C2.2, D1.1, D1.2, D2, and C3. Additionally, to more clearly illustrate the differences between Tieguanyin (*Camellia sinensis* cv. Tieguanyin) and other plants, *Arabidopsis thaliana* was incorporated into the evolutionary analysis. The D1.2 subfamily contains 11 members, making it the subfamily with the largest number of members; the B2 subfamily comprises 3 members, serving as the subfamily with the fewest members. Furthermore, the D1.1 subfamily is the second smallest in terms of the number of members, containing four *CsDOF* genes. In this subfamily, only Tieguanyin has *DOF* gene members, while *Arabidopsis* lacks *DOF* genes. This suggests that the *CsDOF* gene in Tieguanyin may have undergone gene amplification during evolution, leading to the retention of certain genes in Tieguanyin and possibly the acquisition of specific functions unique to Tieguanyin. Homologous genes in common branches may have a similar structure and function [51,52]. Based on previous studies, we found that the *SCAP1* gene plays a key role in regulating the functional differentiation of guard cells, which was identified as *AT5G65590* in *Arabidopsis* [53]. In subfamily B2, *CsDOF14* and *Arabidopsis thaliana AT5G65590.1* (*SCAP1*) show high homology, indicating that *CsDOF14* may play a key role in regulating stomatal guard cell maturation and the terminal stage of guard cell differentiation.

### 2.3. Characterization of the CsDOF Gene Family: Conserved Motifs, Exon–Intron Organization, and Phylogenetic Relationships

To further elucidate the phylogenetic relationships within the *CsDOF* gene family, we constructed a phylogenetic tree based on the amino acid sequences of CsDOF proteins, and combined with correlation analysis, our investigation identified 10 conserved motifs, functional domains, and exon–intron organizational patterns, as illustrated in the accompanying figure (Figure 3). Comprehensive analysis of the *CsDOF* gene family revealed the universal presence of motif 1 across all members (as shown in the motif distribution panel on the left), indicating its crucial role as a core conserved element within the CsDOF functional domain. Comparative analysis of motif organization revealed significant conservation among specific *CsDOF* members, particularly *CsDOF*47, *CsDOF*54, *CsDOF*53, *CsDOF*52, *CsDOF*51, *CsDOF*50, *CsDOF*49, and *CsDOF*48, which exhibited highly similar motif architectures. Phylogenetic and functional analysis suggests that these genes likely share conserved biological functions and may have originated through gene duplication events from a common ancestral gene, thereby maintaining these evolutionarily conserved sequence elements.

The sequence analysis revealed that all 58 identified Tieguanyin DOF proteins contained the characteristic DOF domain, thereby providing conclusive evidence for their classification within the DOF transcription factor family, and structural characterization of the exon–intron organization revealed that 15 DOF proteins exclusively contained the coding sequence (CDS) region, lacking intronic sequences, while the remaining members of the DOF family were characterized by the presence of both untranslated regions (UTRs) and coding sequences (CDSs) in their genomic organization.

### 2.4. Genome-Wide Analysis of CsDOF Genes in Tieguanyin: Uneven Chromosomal Distribution and Duplication-Driven Expansion

To deeply explore the evolutionary connections within the *CsDOF* gene family, phylogenetic analysis was performed (Figure 4). Tieguanyin has 15 chromosomes, with 58 *CsDOF* genes unevenly distributed across 13 chromosomes. Chromosome 8 contained the largest number of *CsDOF* members (13 genes, 22.41%), followed by chromosome 4 (9 genes) and chromosome 15 (6 genes). The remaining 10 chromosomes harbored 1–4 *CsDOF* genes each, while chromosomes 3 and 5 lacked *CsDOF* genes entirely. Intraspecific collinearity analysis revealed 46 paralogous gene pairs unevenly distributed in the Tieguanyin genome, suggesting segmental duplication origins, while 28 paralogous pairs on the same chromosome likely arose from tandem duplication.

Previous studies indicate that gene family expansion primarily occurs through tandem and segmental duplications. This expansion plays critical roles in developmental processes and defense responses [54], potentially enhancing environmental adaptability [55]. Frequent tandem and segmental duplications observed in the Tieguanyin *CsDOF* family suggest that these mechanisms may contribute to improved resistance against adverse conditions.

### 2.5. Cross-Species Collinearity Analysis of DOF Genes in Camellia sinensis and Model Plants Reveals Evolutionary Trajectories and Breeding Implications

This study employed comparative interspecific collinearity analysis of *Camellia sinensis DOF* (*CsDOF*) genes with homologous sequences from *Oryza sativa*, *Solanum tuberosum*, and *Zea mays* to elucidate evolutionary conservation patterns and diversification mechanisms within the *DOF* gene family across major angiosperm lineages. Figure 5 demonstrates conserved syntenic relationships between *Camellia sinensis* cv. Tieguanyin and representative angiosperms: monocots (*Oryza sativa* and *Zea mays*) and the dicot *Solanum tuberosum*. Genomic collinearity analysis identified 19, 101, and 24 homologous gene pairs in rice, potato, and maize respectively, highlighting distinct conservation patterns across divergent angiosperm lineages. Phylogenetic analysis demonstrated a marked increase in homologous gene pairs between *Camellia sinensis* cvs. The 101 collinear pairs with potato represent a 5.3-fold increase compared to rice (19 pairs), conclusively demonstrating stronger synteny conservation within core eudicots. The observed genomic conservation between the two species likely stems from ancestral genetic inheritance, as evidenced by conserved biological processes (including stress adaptation and metabolic homeostasis) and shared architectural features in chromosomal organization. Comparative genomic analyses revealed conserved tandem repeat loci shared among *Camelia sinensis* cv. Tieguanyin, *Oryza sativa*, *Solanum tuberosum*, and *Zea mays*, with these genetic elements demonstrating collinear arrangement across the four phylogenetically diverse species. This investigation yielded dual mechanistic insights, elucidating functional diversification patterns and evolutionary trajectories of DOF transcription factors while establishing a conceptual framework for systematic identification of orthologous DOF regulators in molecular-driven crop breeding programs.

### 2.6. Prediction Analysis of Cis-Acting Elements for CsDOF Gene Families

We analyzed cis-elements in the 2000 bp promoter regions of *CsDOF* genes, identifying five functional categories: hormone response, stress response, light response, tissue/development, and other (Figure 6).

The *CsDOF* promoters contained abundant cis-elements associated with hormone signaling and stress responses. Key hormone-responsive motifs included ABA-, MeJA-, and SA-related elements, suggesting roles in phytohormone regulation. Notably, drought-inducible MYB binding sites were highly enriched in *CsDOF*40, *CsDOF*20, and *CsDOF*44, implicating these genes in drought adaptation. Additionally, heat- and defense-related motifs were identified, linking *CsDOF*s to high-temperature and stress-responsive pathways. This cis-element profiling underscores their potential functional divergence beyond sequence homology, with promoter architecture directly supporting roles in abiotic stress tolerance.

### 2.7. Expression Profile of CsDOF Gene in Roots, Stems, and Leaves of Tieguanyin

In this experiment, three biological replicates were established for the roots, stems, and leaves of Tieguanyin. Subsequently, relevant bioinformatics tools were employed to generate the expression heat map of the *CsDOF* gene (Figure 7). The results were analyzed based on the data from the triplicate experiments. It was found that the expression levels of *CsDOF45, CsDOF31, CsDOF6, CsDOF10*, and *CsDOF14* were relatively low in the roots, stems, and leaves, while the expression levels of *CsDOF27* and *CsDOF28* were higher in roots, stems, and leaves, which indicates that they may play an important role in the growth and development of roots, stems, and leaves. In addition, the experimental findings revealed that the expression levels of *CsDOF32*, *CsDOF34*, *CsDOF35*, *CsDOF27*, and *CsDOF28* were elevated in the leaves. The expression levels of *CsDOF27*, *CsDOF28*, and *CsDOF29* were found to be higher in the roots. The expression levels of *CsDOF35*, *CsDOF27*, and *CsDOF28* were higher in stems. By combining analysis with phylogenetic trees (Figure 2 and Figure 7), we found that most of the genes with increased expression levels belonged to the C2.1 subfamily. Overall, genes from the C2.1 subfamily are highly expressed in the roots, stems, and leaves of Tieguanyin plants. This suggests that the C2.1 subfamily is primarily responsible for regulating the growth and development of the plant’s roots, stems, and leaves. In contrast, the B2 and C3 subfamily genes have lower overall expression levels. However, this does not mean these genes are unimportant; they may also be expressed in other parts of the plant.

### 2.8. Predicted Protein Secondary and Tertiary Structures of the CsDOF Genes

Secondary structure analysis reveals that the *DOF* gene family encodes proteins with the highest proportion of irregular coils, followed by α-helices and extended strands. No β-sheet or β-turn characteristics were observed, except for *CsDOF*31, which exhibited a relatively low α-helix content. Previous studies have demonstrated that protein secondary structure stability primarily depends on α-helices [56]. These findings suggest that the *DOF* gene family exhibits lower structural stability, consistent with the instability coefficients derived from physicochemical property analysis.

There is a close relationship between the structures of proteins and their functions, the secondary, tertiary, and quaternary structures being crucial for forming the unique three-dimensional conformation of a protein. According to the analysis of the tertiary structure diagram (Figure 8), the members of the *CsDOF* gene family lack similar tertiary structures, further demonstrating structural differences among the subfamilies within the *CsDOF* gene family and highlighting the diversity of CsDOF protein functions.

### 2.9. Expression Analysis of DOF Family Genes in Tieguanyin Under Drought and Heat Treatments

According to the phylogenetic tree (Figure 2), it can be seen that the B1 subgroup is involved in ABA regulation, and ABA is related to stress [57]; the D1 subgroup is rich in stress-related genes. To further investigate the functions of the *CsDOF* gene and its expression, two abiotic stresses (high temperature: 40 °C, drought: 10% PEG 6000) were used as research subjects. Tea leaves were collected at 0 h, 4 h, 8 h, 12 h, and 24 h post-treatment, and qRT-PCR was employed for detection. Among these, 13 representative *CsDOF* genes were identified in subgroups B1 and D1 (Figure 9).

When the plants were under high-temperature stress, we observed that most *CsDOF* expression patterns showed “rise first and then decline”. *CsDOF45*, *CsDOF12*, *CsDOF9*, and *CsDOF43* reached the peak expression level at 12 h and then decreased, while *CsDOF46* reached the peak expression level at 8 h and then decreased. The expression pattern of a few genes (*CsDOF51* and *CsDOF52*) was opposite. However, some notable exceptions were observed. For instance, the expression patterns of *CsDOF10* and *CsDOF42* showed a “rise, fall, and rise again” pattern, with turning points at 8 h and 12 h post-treatment. *CsDOF7*, *CsDOF11*, and *CsDOF8* exhibited slightly reduced expression from 0 to 4 h, but unlike *CsDOF7*, the other two *CsDOF* genes reached their peak expression levels at 12 h and then declined. In contrast, *CsDOF7* was more susceptible to high-temperature stress, showing greater fluctuations in expression. After multiple comparisons and ANOVA, it was found that both drought and heat stress significantly affected the expression of the *CsDOF* gene, but the expression of *CsDOF* under heat stress was more pronounced than under drought stress (*p* < 0.05). *CsDOF51* and *CsDOF44* showed a more significant response to drought stress, with their expression levels reaching approximately 3.5 times and 3 times those of the control group after 24 h, respectively.

In conclusion, *CsDOF* gene expression changed in different periods under drought and heat treatment, indicating that *CsDOF* may play a key role in coping with plant stress. The highest expression level of *CsDOF51* was higher than that under heat stress (*p* < 0.05), while the highest expression level of *CsDOF44* and *CsDOF12* was significantly higher than that under drought stress (*p* < 0.05). It is speculated that the *DOF* gene of Tieguanyin may mainly respond to drought stress by *CsDOF51* and high-temperature stress by *CsDOF12* and that *CsDOF44* responds to both stresses.

## 3. Discussion

Tieguanyin (*Camellia sinensis* cv. Tieguanyin) is a premium oolong tea cultivar prized for its distinctive flavor and aroma. Its exceptional quality grants it a high market value and significant research importance in tea science. Due to the increase in extreme weather, greenhouse effects, and carbon dioxide concentrations, the carbon–nitrogen balance is upset, and the growth, development, and quality of plants are significantly affected [58]. Glutamine, as a key amino acid in nitrogen metabolism, is not only involved in regulating plant quality but also in regulating plant abiotic stress response [59]. In tea plants, the *CsDOF* gene is closely related to the biosynthesis of glutamine, which can support nitrogen mobilization from mature leaves to new buds and improve the quality of tea [60,61]. Comparative genomic analyses reveal that tea plants have experienced dynamic gene family contraction and expansion events, conferring distinctive adaptive advantages for thriving in montane ecosystems [62]. Under environmental stress conditions, plants activate intricate signaling cascades in which DOF transcription factors serve as pivotal regulators, modulating downstream gene networks to mediate abiotic stress tolerance responses [24]. Molecular studies have established DNA-binding with one finger (DOF) transcription factors as essential regulatory elements that orchestrate crucial physiological processes in plants, including growth and development, metabolic regulation, and environmental stress responses. Therefore, they can help plants overcome different stresses and optimize their growth. The study of DOF protein in Tieguanyin and its mechanism of action can lay a foundation and provide ideas for improving the drought resistance and heat resistance of Tieguanyin in the future. Meanwhile, plants achieve a multi-level regulatory network for environmental stress through the redundancy and functional differentiation of gene family members, a strategy that is particularly important in the adaptation to extreme climates. This evolutionary strategy proves particularly crucial for extreme climate adaptation, where functional redundancy among paralogous genes ensures robust stress response systems [63].

Thus, we characterized the physicochemical properties of proteins encoded by the *CsDOF* gene family, including predictions of amino acid length, molecular weight, isoelectric point (pI), instability index, aliphatic amino acid index, and hydrophilicity/hydrophobicity. These predictive analyses furnish crucial insights for a more in-depth comprehension of the physical characteristics of CsDOF proteins. The physicochemical property prediction of CsDOF proteins indicated their general hydrophilicity and structural instability, suggesting that they may dynamically respond to environmental signals through a rapid turnover mechanism. The hydrophilic and unstable characteristics of CsDOF proteins are beneficial for their rapid response to environmental stress, which is of great significance for the adaptation of tea plants to the external environment [64]. The results demonstrated that the majority of CsDOF proteins exhibited structural instability, rendering them susceptible to proteolytic degradation, and possessed hydrophilic properties. These findings are congruent with the overall hydrophilic characteristics of the *DOF* gene family.

Additionally, we performed a systematic analysis of the *DOF* gene family in the Tieguanyin tea plant (*Camellia sinensis* cv.) and successfully identified 58 genes belonging to this family. These genes are distributed across 15 chromosomes, exhibiting uneven distribution patterns, and a gene clustering phenomenon was observed. As observed in the tea tree *DOF* (*CsDOF*) gene family, the clustered distribution of genes on chromosomes is frequently linked to coordinated regulatory functions and evolutionary innovations. In the analysis of the chromosome distribution of *CsDOF* genes in Tieguanyin (*Camellia sinensis*), tandem repeats and segmental repeats were also observed. Tandem and segmental duplications serve as the primary evolutionary forces driving gene family expansion. These genomic events facilitate functional diversification and enhance organismal adaptation to environmental stresses [65]. Subsequent studies demonstrated that the *DOF* genes in Tieguanyin underwent continuous expansion via tandem and segmental duplications. This amplification process provides the genetic basis for gene functional divergence, endowing Tieguanyin (*Camellia sinensis*) with more sophisticated and diversified regulatory mechanisms in growth, development, and environmental responses. Consequently, it enhances the plant’s resistance to both biotic and abiotic stresses, significantly improving its environmental adaptability. Simultaneously, this process induced alterations in the genomic architecture, exemplifying the dynamic evolutionary traits of the Tieguanyin genome. It thereby propels the continuous evolution of the Tieguanyin cultivar.

A phylogenetic tree is usually used to illustrate the relationships between different species. We constructed a phylogenetic tree by comparing the sequences of DOF proteins in *Arabidopsis thaliana* and Tieguanyin (*Camellia sinensis* cv.). The analysis shows that the homologous genes of Tieguanyin and *Arabidopsis thaliana* are divided into four major families (A, B, C, and D), which is consistent with previous studies [66]. In earlier studies, these genes were further categorized into seven or nine subfamilies. However, this study included more genes, leading to an increase in the number of subfamilies. Based on previous classifications, ten subfamilies were ultimately identified: Subgroup A, Subgroup B1, Subgroup B2, Subgroup C1, Subgroup C2.1, Subgroup C2.2, Subgroup C3, Subgroup D1.1, Subgroup D1.2, and Subgroup D2 [66,67,68]. The phylogenetic tree indicated that *CsDOF14* shared a high level of sequence homology with *Arabidopsis thaliana AT5G65590.1* (*SCAP1*), implying that *CsDOF14* may exhibit biological functions analogous to those of *SCAP1*. Genes, particularly those coding for terpene biosynthesis proteins, associated with tea aroma and stress resistance were significantly amplified through recent tandem duplications and exist as gene clusters in the tea plant genome [64].

Synteny analysis revealed conserved genomic blocks containing *DOF* genes across *Camellia sinensis*, *Arabidopsis thaliana*, and *Oryza sativa*, supporting their evolutionary origin from a shared ancestral genome in core eudicots [64]. To elucidate the evolutionary relationships between *Camellia sinensis* cv. Tieguanyin and three representative angiosperms (*Oryza sativa*, *Solanum tuberosum*, and *Zea mays*), we conducted comparative synteny analysis of their orthologous *DOF* gene sequences. Comparative genomic analysis revealed the highest abundance of syntenic *DOF* gene pairs between *Camellia sinensis* cv. Tieguanyin and *Solanum tuberosum*, suggesting their relatively closer phylogenetic relationship among the analyzed species. Comparative genomic analysis revealed that *Camellia sinensis* cv. Tieguanyin shares orthologous *DOF* genes with *Oryza sativa* (rice), *Solanum tuberosum* (potato), and *Zea mays* (corn). This conservation pattern not only highlights the functional importance of the *DOF* gene family throughout plant evolution but also supports its derivation from a common ancestral gene in angiosperms.

In the analysis of intron–exon sequences, the *CsDOF* gene exhibited a highly conserved zf-DOF domain, indicating that the evolutionary trajectory of the *DOF* family in Tieguanyin is relatively conservative. Most *CsDOF* genes within the same subfamily show similar motif structures, suggesting they share similar functions in plant growth, development, and stress response. Phylogenetic and functional analyses indicate that these genes may share conserved biological functions and could have originated from a common ancestral gene through a gene duplication event, thus maintaining these evolutionarily conserved sequence elements. The number of introns in the *CsDOF* gene of Tieguanyin is between zero and two. Studies have shown that the presence of introns is beneficial to the function of organisms [69]. However, intron-free genes can also respond quickly to external stimuli by bypassing the splicing process, which is crucial for organisms to adapt to stress conditions [70].

Analysis of tissue-specific expression revealed that genes from different subfamilies exhibit varying levels of expression in roots, stems, and leaves. Similarly, multiple plant gene families have shown similar tissue-specific expression patterns [71]. Further analysis revealed that among these subfamilies, the C2.1 subfamily has the highest proportion of highly expressed genes in the roots, stems, and leaves of Tieguanyin, strongly indicating its crucial role in regulating root, stem, and leaf growth. Previous studies have found that *AtDAG1*, which regulates seed germination, is also part of the C2.1 subfamily, suggesting that most of its genes may play a significant role in regulating seed germination and crop growth [66,72]. Root exudates play a crucial role in plant–microbe interactions, and genes related to root exudation in tea plants are likely involved in regulating the recruitment of beneficial rhizosphere microorganisms, thereby affecting nutrient uptake and plant stress resistance [73]. In sharp contrast, B2 and C3 subfamily genes exhibit relatively low overall expression levels in roots, stems, and leaves, implying their potential critical functions in other tissues and organs.

In conclusion, as elaborated in the study “Genome-wide identification of the DNA-binding one zinc finger (*DOF*) transcription factor gene family and their potential functioning in nitrogen remobilization in tea plant (*Camellia sinensis* L.)”, this study conducted a multi-dimensional and comprehensive investigation of the *DOF* gene family in Tieguanyin tea plants. This study comprehensively and systematically uncovered various characteristics of the gene family, covering gene identification, chromosomal distribution, evolutionary analysis, protein physicochemical property studies, interspecific relationships, and gene expression profiling in different tissues [74]. These findings not only augment our comprehension of the genetic underpinnings in Tieguanyin tea (*Camellia sinensis*) but also furnish a robust theoretical foundation for future molecular breeding efforts. Such efforts aim to enhance varietal resilience to environmental stresses, thereby safeguarding the quality and yield of Tieguanyin tea. On this foundation, subsequent research could prioritize elucidating the precise mechanisms by which C2.1 subfamily genes regulate root, stem, and leaf growth. Additionally, functional analyses of B2 and C3 subfamily genes in other tissues and organs are warranted. Simultaneously, in-depth exploration of the detailed molecular pathways underlying the rapid cellular signal responses of the 15 unique genes is essential. These efforts will steadily broaden our understanding of tea biology and invigorate the sustainable development of the tea industry.

It has been demonstrated that the *DOF* gene responds to different types of abiotic stress, e.g., overexpression of *AcCDF4* promotes drought tolerance in *Arabidopsis thaliana* and overexpression of *SlCDF1* and *SlCDF3* leads to increased salt and drought tolerance in transgenic *Arabidopsis thaliana*. In this study, we investigated the expression patterns of *CsDOF* under drought and high-temperature stress [75,76]. Most *CsDOF* genes showed different degrees of response to drought and high-temperature stress, indicating that *CsDOF* genes may play various roles in drought and high-temperature stress. Under drought stress, *CsDOF51* and *CsDOF44* showed significant responses to drought stress; under heat stress, *CsDOF44* and *CsDOF12* exhibited the most significant responses, with these genes located in the D1 and B1 subfamilies. Previous studies have identified *CsDOF-22*, a gene that plays a role in resistance to osmotic stress and drought stress, in the D2 subfamily [67]. Based on the shared gene *AT3G47500.1*, further comparison showed that the D2 subfamily in this study was consistent with our D1.2 subfamily. Given that genes in the same subfamily may share similar functions, this suggests that the D1.2 subfamily may play a crucial role in responding to drought stress [77]. These results suggest that the *CsDOF* gene family plays an important role in the resistance of Tieguanyin to drought and high-temperature stress and promotes its recovery after resistance. The results provide a basis for studying the *CsDOF* gene in the stress response mechanism of Tieguanyin and also provide new insights for further improving the drought and heat tolerance of Tieguanyin.

Under high-temperature and drought stress, the water supply to the plant rhizosphere will be reduced and the absorption of nutrients by the root system will be inhibited [78]; excessive expression of reactive oxygen species (ROS) will destroy the permeability of membranes [79,80], leading to oxidative stress [81], and will also cause stomatal closure, a decrease in chlorophyll content, inhibition of photosynthesis [82,83], etc. Studies have shown that the *DOF* gene is involved in the response of plants to drought and high-temperature stress. However, the exact mechanism behind this process is still poorly understood. Under drought and heat stress, *DOF* gene expression promotes the coding of heat shock protein (HSF), the synthesis of peroxidase (ASA, CAT, etc.), and the promotion of root and bud growth, which maintain intracellular ROS homeostasis and respond to adversity from plant morphology [84]. When plants are subjected to stress stimulation, the channels are activated instantly, and Ca2+ flows inward to regulate cytoplasmic Ca2+ concentrations. Through the induction of related protein kinases (such as CaM and CDPK) [85], a series of biochemical reactions are induced to adapt or resist various stresses [86]. Mitogen-activated protein kinase (MAPK) can participate in the signal transduction during the response of eukaryotes to extracellular stimulation [87]. When Ca2+ flows in, it induces the activation of Ca2+-dependent protein kinase (CDPK), which further activates heat-stressed HAMK (heat-shock-activated MAPK) [87]. According to previous studies, the *DOF* gene can participate in the MAPK cascade reaction to respond to environmental stress [88]. In addition, heat-activated MAPK is also involved in the expression of *HSP* genes [87]. That is, under heat stress, both the MAPK signaling pathway and the HSF signaling pathway are activated [79]. When ROS is accumulated in excess, the activity of antioxidant enzymes such as superoxide dismutase, catalase, ascorbic acid peroxidase and peroxidation enzyme in plants will be enhanced to start the mechanism of ROS removal [89]. Moreover, HsfA4a in heat-induced transcription factor (HSF) can also act as an exogenous ROS sensor. When it senses ROS signals, it transfers ROS to downstream transcription factors through the MAPK signaling pathway and reduces ROS levels under the joint action of downstream transcription factors (the *MAPK* gene family and the *Zat* gene family) and antioxidant enzymes [88]. In addition to signaling pathways, plants can enhance their resistance to stress by altering their morphology. Under drought and high temperatures, plants typically increase root length and weight to access deeper soil moisture under drought stress [90], and they also increase leaf movement and stem–leaf epidermal waxes to reduce incoming solar radiation, thereby minimizing water loss [91,92,93]. In the study of cis-acting elements (Figure 6) and plant tissue expression (Figure 7), we found that *CsDOF* contains an auxin-responsive element and exhibits differential expression among different vegetative tissues through *DOF*. We speculate that these *DOF* genes (*CsDOF27*, *CsDOF28*, *CsDOF36*, *CsDOF37*, *CsDOF56*, *CsDOF9*, *CsDOF17*, and *CsDOF18*) play a crucial role in promoting root growth and plant morphological changes under drought and high-temperature conditions. In the entire process of plants resisting stress, plant hormones also play a crucial role. Under high temperatures and drought conditions, plant hormones such as SA, MeJA, and ABA interact within the plant body. Not only do they respond to plant defense mechanisms, enhancing drought and heat tolerance, but they also actively contribute to the clearance of ROS [94,95,96,97,98]. In the analysis of cis-acting elements (Figure 6), we found that the *CsDOF* gene has a potential relationship with these plant hormone elements, indicating that the *CsDOF* gene plays a significant role in resisting adverse environments (Figure 10).

This mechanism is an assumption derived from our study of the common response of *CsDOF* and other plants to stress. Further experiments are needed to confirm whether it indeed helps plants overcome adverse challenges by regulating the expression of these proteins. Tea is rich in polyphenols and ascorbic acid (AsA) [99]. The content of AsA in tea is not only related to its flavor but also to its storability [99]. Studies have shown that the *DOF* gene is also involved in AsA expression [100]. Our research aims to enhance the drought and heat tolerance of Tieguanyin and provide a basis for strategies to improve tea quality and flavor.

## 4. Materials and Methods

### 4.1. Data Sources

The genomic files and annotation files of Tieguanyin (*Camellia sinensis* cv.) were sequenced in Shenzhen. The assembly and annotation were archived in the National Center for Biotechnology Information under the accession number JAFLEL000000000 and in the GWH (https://bigd.big.ac.cn/gwh/, accessed on 3 April 2024) under accession numbers GWHASIV00000000 for the monoploid and GWHASIX00000000 for the haplotype-resolved genome. The genomic sequences and annotation files of rice (*Oryza sativa* L.), potato (*Solanum tuberosum* L.), and corn (*Zea mays* L.) were downloaded from the Ensembl Plants database (https://plants.ensembl.org/index.html, accessed on 3 October 2024) by selecting the corresponding plant’s FASTA and GFF3 files and clicking for download. The genomic sequences and annotation files of *Arabidopsis thaliana* (L.) Heynh were downloaded from the TAIR database (https://www.arabidopsis.org/, accessed on 3 January 2024). The protein sequences of the *DOF* transcription factor in *Arabidopsis thaliana* (L.) Heynh were downloaded from the PlantTFDB (https://planttfdb.gao-lab.org/, accessed on 15 April 2025) by selecting the *DOF* option, locating Arabidopsis thaliana, and clicking “Download Sequences” to obtain the Arabidopsis *DOF* protein sequences. The HMM file (PF02701) for the *DOF* (DNA-binding One Zinc Finger) domain was obtained from the Pfam database (http://pfam.xfam.org/, accessed on 3 October 2024).

### 4.2. Identification and Physicochemical Properties of DOF Gene Family

Using the TBtools’s (version 2.310) Blast and Hmmsearch modules, with *Arabidopsis* as the reference gene and Pf02701′s HMM model, we searched the entire genome sequence of Tieguanyin and took the intersection. For candidate sequences, we used NCBI (https://www.ncbi.nlm.nih.gov/Structure/bwrpsb/bwrpsb.cgi, accessed on 3 April 2024) online tools to confirm the integrity of *DOF* domains and performed multi-sequence alignment of DOF proteins using DNAMAN software (version 9.0), removing unreasonable sequences. The amino acid sequences of CsDOF proteins were uploaded to the online website Expasy-ProtParam (https://web.expasy.org/protparam/, accessed on 3 April 2024) for physicochemical property analysis, including the number of amino acids corresponding to each protein, molecular mass, isoelectric point, instability index, and other information, which was statistically analyzed by WPS excel and made into a table.

### 4.3. Analysis of Gene Structure and Conserved Motifs

Based on the gene annotation files (gff3/gtf), we used the Gene Structure View module of TBtools to visualize the exon–intron distribution. Conserved motifs were predicted using the MEME online tool and TBtools (the motif count was set to 10). The protein domain composition was further analyzed through TBtools’s Visualize NCBl CDD Domain Pattern.

### 4.4. Chromosome Mapping and Collinearity Analysis

Using the Gene Location Visualize from GTF/GFF function of TBtools, the *DOF* gene was mapped to the chromosome (the input file was gff3/gtf, and the chromosome length and gene position parameters were adjusted). The MCScanX tool was used to detect the duplicated gene pairs in the genome of Tieguanyin, and the Circos Plot of TBtools was used to visualize the collinearity. The *DOF* genes of Tieguanyin (*Camellia sinensis* var. sinensis), *Arabidopsis thaliana* (*Arabidopsis thaliana*), rice (*Oryza sativa* L.), potato (*Solanum tuberosum* L.), and corn (*Zea mays*) were used as references, and cross-species collinearity was analyzed by MCScanX.

### 4.5. Phylogenetic Tree Construction

Based on the sequence files of *DOF* family genes in Tieguanyin and *Arabidopsis thaliana*, a phylogenetic tree was constructed using the One Step Build a ML Tree module in TBtools through the neighbor-joining method (NJ), and 1000 self-copies were calculated. Finally, the phylogenetic tree was polished using Adobe Illustrator CC 2018.

### 4.6. Promoter Analysis

The Sequences Extract module in TBtools was used to obtain the upstream 2000 bp of each *CsDOF* gene as a promoter. The cis-elements in the *CsDOF* promoter regions were obtained through PlantCARE (https://bioinformatics.psb.ugent.be/webtools/plantcare/html/, accessed on 20 April 2024) and visualized using the Simple Biosequence Viewer module in TBtools.

### 4.7. Expression Pattern Analysis (Heat Map Construction)

Based on FPKM (fragments per kilobase of transcript per million mapped reads) values, the heat maps of Tieguanyin *DOF* genes in roots, stems, and leaves were constructed. Roots were sampled from young lateral roots (non-lignified), stems from the middle segments of primary shoots, and leaves from fully expanded young leaves. The heat maps were created using the HeatMap module in TBtools. Finally, TBtools was used to refine the heat maps.

### 4.8. Protein Structure Prediction

SOPMA was used to predict the proportions of α-helices, extended chains, and irregular curls. Three-dimensional structures of CsDOF proteins were modeled via Swiss-Model (https://swissmodel.expasy.org/, accessed on 20 April 2024) using *Camellia sinensis DOF* FASTA sequences, and the structural differences between subfamilies were analyzed.

### 4.9. Sources of Plant Materials and Abiotic Stress Treatments

Camellia sinensis (cv. Tieguanyin) seedlings were sourced from Xiping Town, Anxi County, Fujian Province. These seedlings were cultivated under multiple stress treatments in an artificial climate chamber. Samples of Tieguanyin were collected and stored in liquid nitrogen at −80 °C for subsequent RNA extraction.

Twenty Tieguanyin seedlings were selected, and their roots were rinsed with distilled water. The treatment group was immersed in a nutrient solution containing 10% polyethylene glycol 6000 (PEG6000) and cultivated in an artificial climate chamber set at 25 °C and 75% relative humidity. Samples were collected from the treatment group at the specified time points, while control samples were harvested at 0 h. The control group was maintained at room temperature, whereas 20 seedlings in the treatment group were placed in a light incubator with a treatment temperature of 40 °C and 75% relative humidity. Treatment durations were set as 4 h, 8 h, 12 h, and 24 h. Control samples were collected at 0 h for baseline comparison.

### 4.10. RNA Extraction and qRT-PCR Analysis

Total RNA was extracted using an RNA extraction kit (Beijing Biomed Gene Technology Co., Ltd., Beijing, China, catalog number RA106-01). Construction of the cDNA library was performed using Biomed’s MT403-01 kit. Specific primers for target genes were designed in their non-conserved regions using TBtools software (version 2.210) and synthesized by Bosham Biotechnology Co., Ltd., Fuzhou, China.

For real-time fluorescent quantitative analysis, the reaction program was set as follows: pre-denaturation at 95 °C for 30 s; cycling stage (40 cycles) consisting of denaturation at 95 °C for 5 s and annealing/extension at 60 °C for 30 s; melting curve analysis, including 95 °C for 5 s, 60 °C for 60 s, and 50 °C for 30 s. The internal reference gene was CsGAPDH (accession number GE651107). The relative expression levels of target genes were calculated using the 2^−ΔΔCt^ method. Quantitative data were statistically analyzed and visualized using GraphPad Prism10.4.1 software.

## 5. Conclusions

The *DOF* transcription factor family is critically involved in regulating plant growth and development, metabolic processes, and stress responses. In this study, 58 *CsDOF* genes were identified in the Tieguanyin tea plant, which were phylogenetically classified into nine distinct subfamilies. A comprehensive analysis was conducted, including gene localization, physicochemical characterization, conserved motifs, exon–intron structures, and tissue-specific expression profiles (roots, stems, and leaves). Additionally, intraspecific and interspecific collinearity, protein structural prediction, and differential expression patterns under drought, high-temperature, and high-light stress conditions were systematically investigated. These findings provide valuable insights into the molecular mechanisms underlying *CsDOF*-mediated stress responses to high temperatures and drought in Tieguanyin tea plants.

This study establishes a foundation for further functional characterization of *CsDOF* genes in Tieguanyin tea plants while providing a theoretical framework for investigating the regulatory mechanisms of *CsDOF* transcription factors in abiotic stress responses. Given the increasing severity of global warming and drought conditions, advancing research on Tieguanyin cultivar improvement will facilitate the expansion of its cultivation range.

## Figures and Tables

**Figure 1 plants-14-01829-f001:**
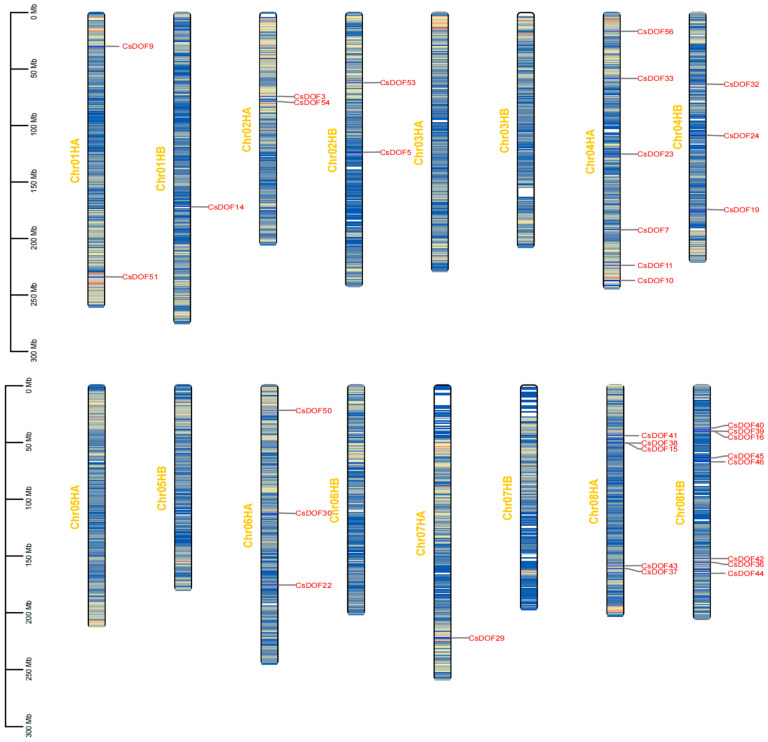
Distribution of *CsDOF* genes in *Camellia sinensis* cv. Tie-guanyin chromosomes. The chromosomal map displays gene density gradients (blue: <5 genes/10 Mb, yellow: >15 genes/10 Mb, white: no genes) across 15 chromosomes (Chr01HA-Chr15HB). Chromosome IDs are annotated on the left, with the physical length scale (0–300 Mb) indicated at the left margin. HA and HB correspond to haploid chromosomes of the paternal and maternal parents, respectively. Gene clusters are particularly enriched in Chr08HA/08HB (yellow sectors).

**Figure 2 plants-14-01829-f002:**
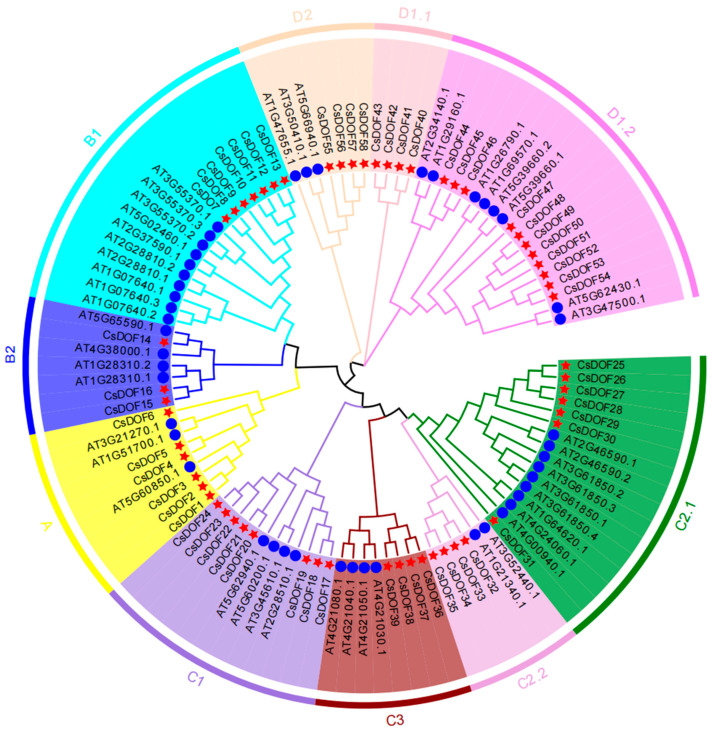
Phylogenetic classification and functional divergence of the *DOF* gene family in *Arabidopsis thaliana* and *Camellia sinensis*: insights from nine evolutionarily distinct subgroups. Forty-seven AtDof proteins were labeled with blue circles, and 58 CsDof proteins were labeled with red five-pointed stars. The obtained phylogenetic trees are divided into four major categories (A, B, C and D) and ten subcategories (A, B1, B2, C1, C2.1, C2.2, D1.1, D1.2, D2, and C3). Different colors on the phylogenetic tree represent different subfamilies.

**Figure 3 plants-14-01829-f003:**
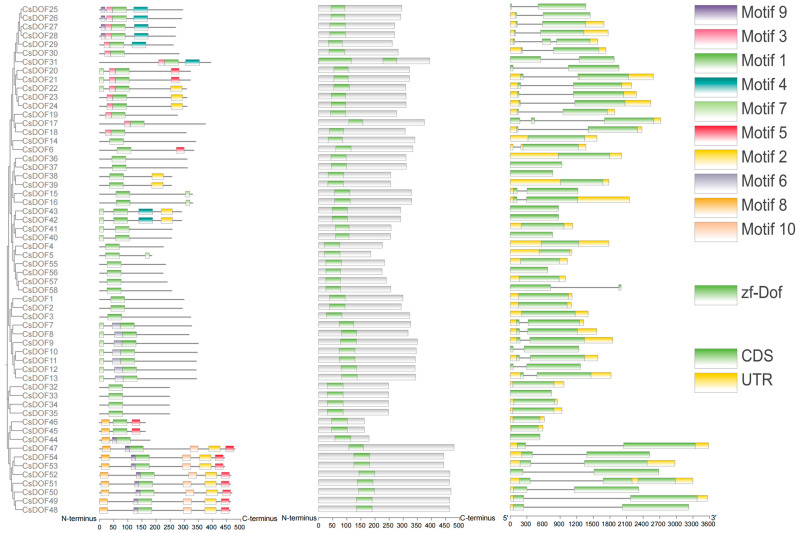
Comprehensive analysis of the *DOF* gene family in Tieguanyin. The first column shows the conserved motifs in CsDOF proteins. The second column shows the analysis of conserved domains in CsDOF proteins. The third column shows the structures of the introns and exons of Tieguanyin *CsDOF* genes. The green boxes represent the exons, the black lines represent the introns, and the yellow boxes represent the untranslated regions (UTRs) upstream and downstream.

**Figure 4 plants-14-01829-f004:**
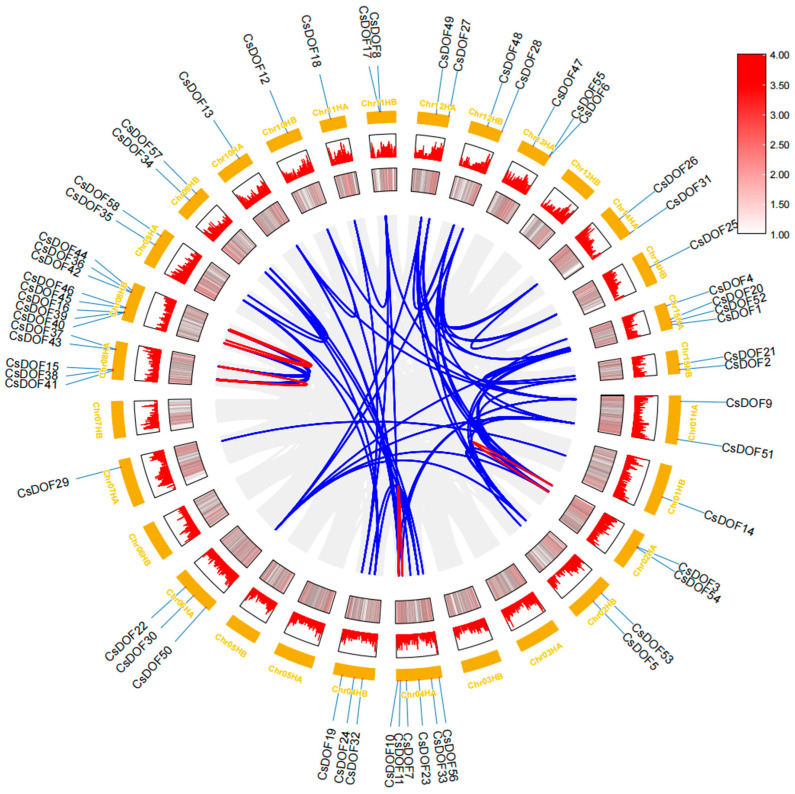
Circso diagram of *CsDOF* genes. The external to internal heat map illustrates how gene densities and GC ratios are distributed along the chromosome. Syntenic segments in the tieguanyin genome are denoted by gray lines, while black lines represent pairs of *CsDOF* genes that are collinear.

**Figure 5 plants-14-01829-f005:**
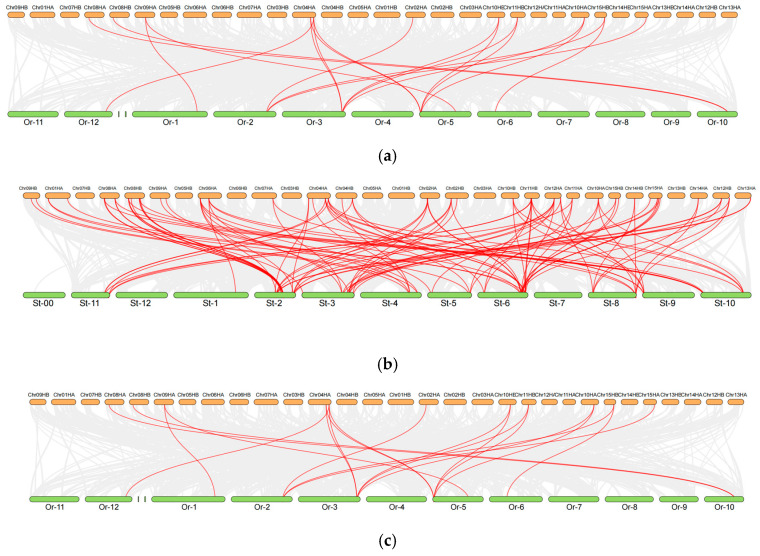
(**a**) Synteny analysis of *DOF* genes between *Camellia sinensis* Cv. Tieguanyin and *Oryza sativa*. (**b**) Synteny analysis of *DOF* genes between *Camellia sinensis* cv. Tieguanyin and *Solanum tuberosum*. (**c**) Synteny analysis of *DOF* genes between *Camelia sinensis* cv. Tieguanyin and *Zea mays.* Different species names and chromosomes are represented by different colors. The red lines represent the homologous *DOF* gene pairs between other species and the Tieguanyin *DOF* genes (*CsDOFs*), and the gray lines represent all homologous gene pairs on the chromosome.

**Figure 6 plants-14-01829-f006:**
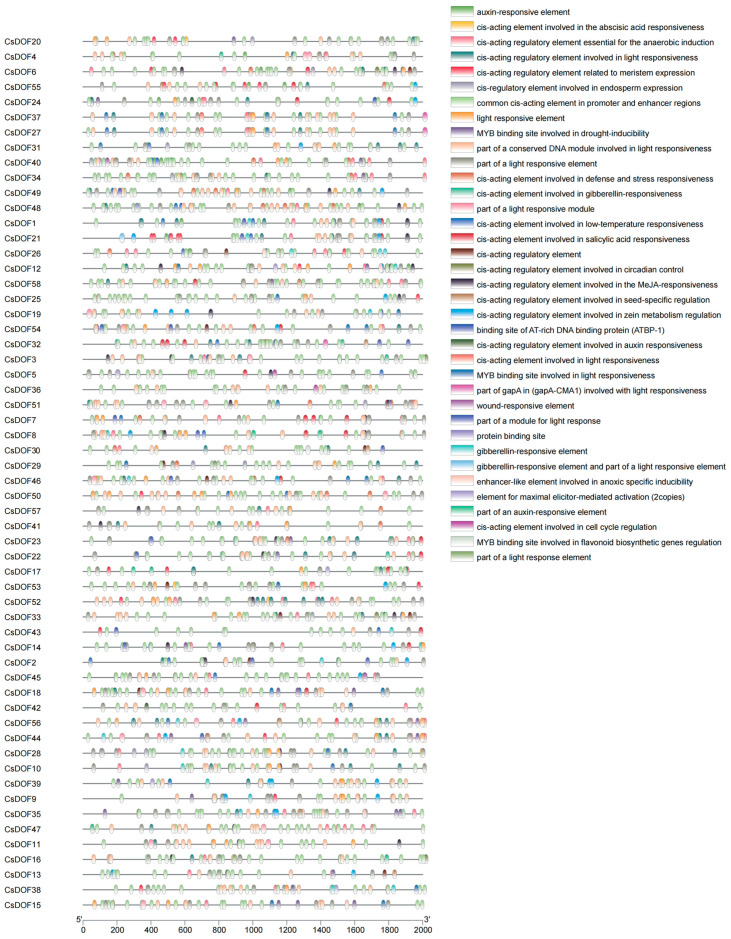
Distribution of cis-elements in the 2000 bp promoter region upstream of different *CsDOF* genes. The legend on the right corresponds to the element types.

**Figure 7 plants-14-01829-f007:**
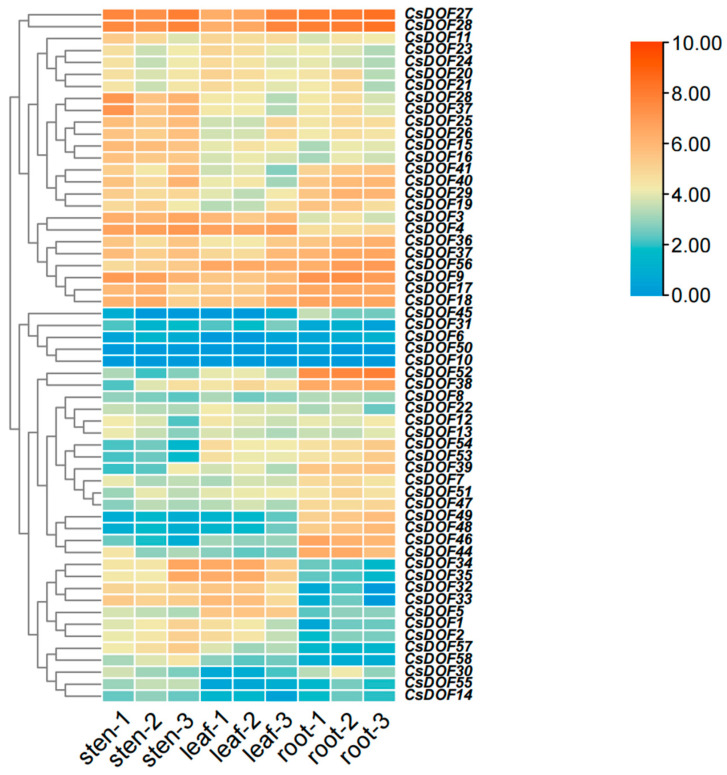
Relative expression levels of *CsDOF* gene in roots, stems, and leaves. Expression profile of *CsDOF* gene in Tieguanyin roots, stems, and leaves. Red means high gene expression in an organ, and blue means low expression. The color scale is on the right of the chart.

**Figure 8 plants-14-01829-f008:**
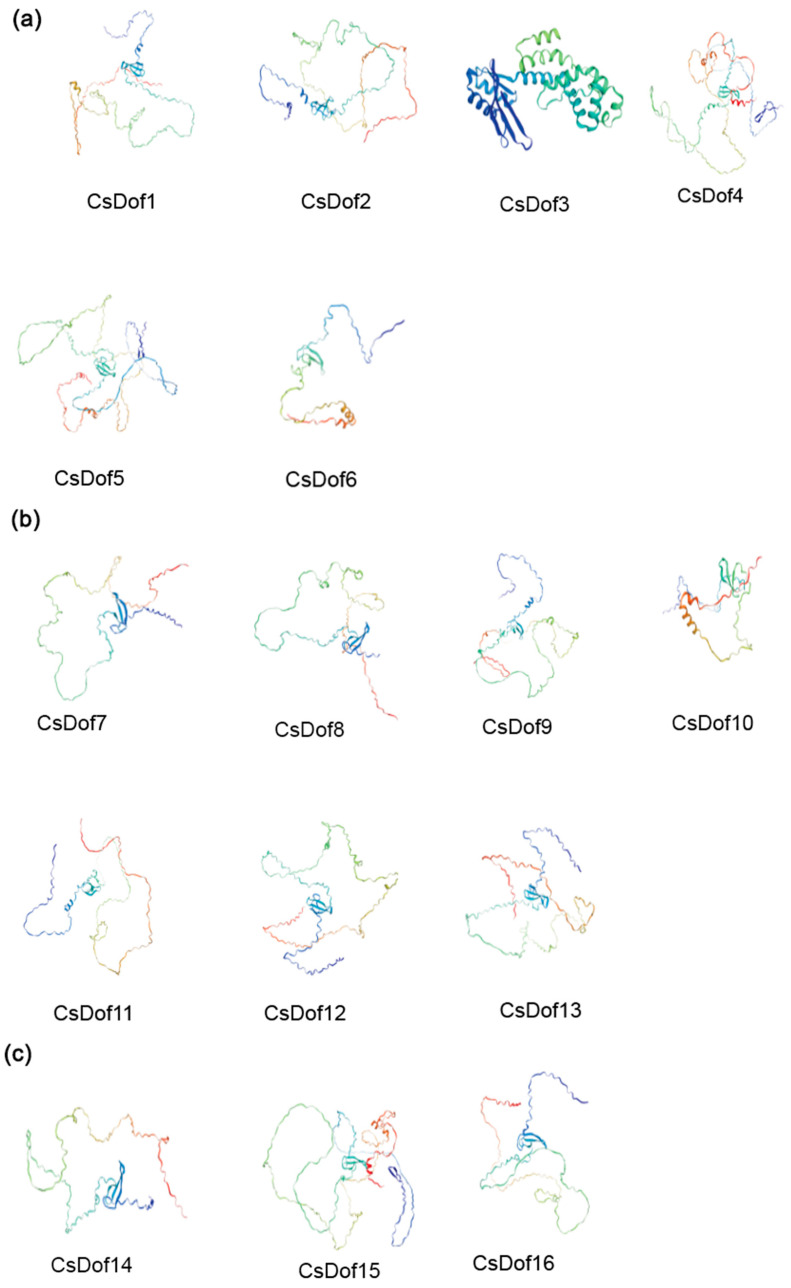
The tertiary structure of members of the CsDOF family. (**a**) Refers to the tertiary structure of CsDOF family members in Subfamily A. (**b**) Refers to the tertiary structure of CsDOF family members in Subfamily B1. (**c**) Refers to the tertiary structure of CsDOF family members in Subfamily B2. (**d**) Refers to the tertiary structure of CsDOF family members in Subfamily C1. (**e**) Refers to the tertiary structure of CsDOF family members in Subfamily C2.1. (**f**) Refers to the tertiary structure of CsDOF family members in Subfamily C2.2. (**g**) Refers to the tertiary structure of CsDOF family members in Subfamily C3. (**h**) Refers to the tertiary structure of CsDOF family members in Subfamily D1.1. (**i**) Refers to the tertiary structure of CsDOF family members in Subfamily D1.2. (**j**) Refers to the tertiary structure of CsDOF family members in Subfamily D2.

**Figure 9 plants-14-01829-f009:**
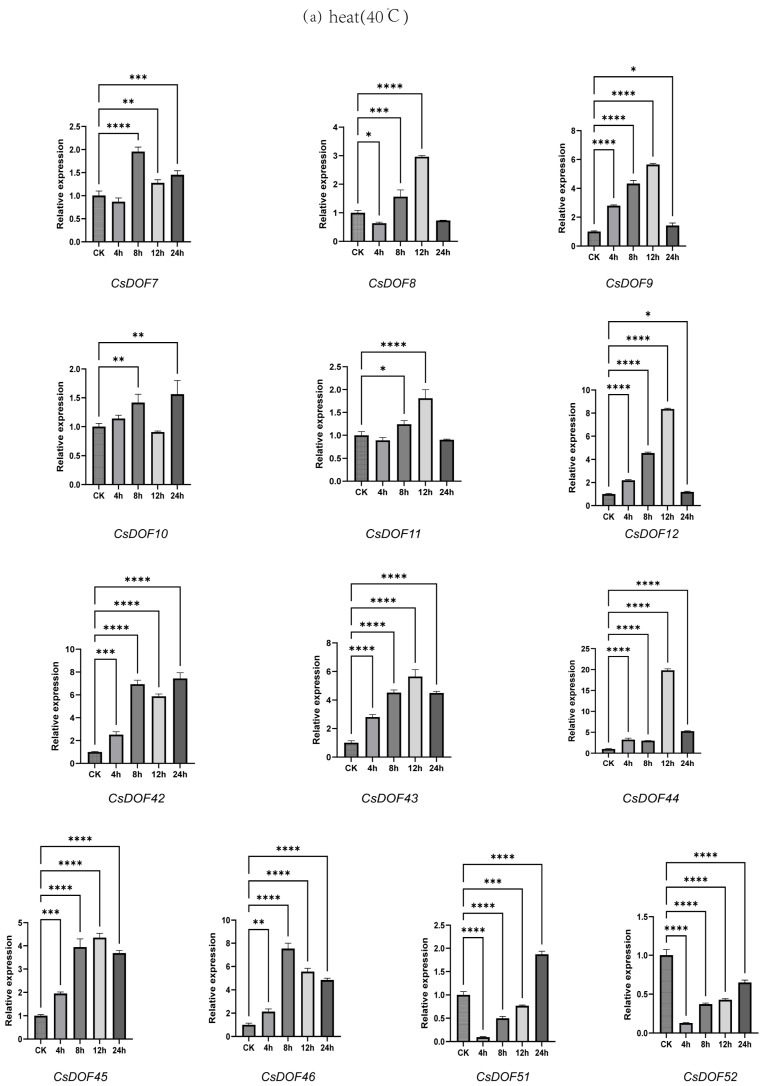
The real-time quantitative PCR (RT-qPCR) expression levels of the selected *CsDOF* gene under high-temperature treatment at 40 °C (**a**) and drought at 10%PEG (**b**). RT-qPCR was used to analyze the expression levels of the *CsDOF* genes. The experimental sample size was 3. Statistical analysis used one-way ANOVA to determine significant differences, with the number of “*” indicating the level of significance (* *p* ≤ 0.05; ** *p* ≤ 0.005; *** *p* ≤ 0.0005; **** *p* ≤ 0.0001).

**Figure 10 plants-14-01829-f010:**
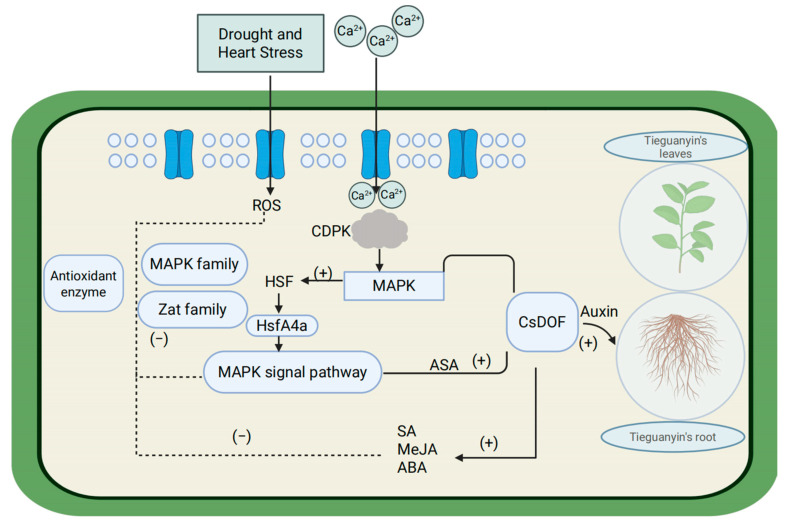
*CsDOF* predictions of the response mechanisms to drought and high-temperature stress. HSF: heat shock protein, CDPK: calcium-dependent protein kinase, MAPK: mitogen-activated protein kinase, HsfA4a: member of the heat shock protein family, AsA: ascorbate, SA: salicylic acid, MeJA: methyl jasmonate, ROS: reactive oxygen species.

**Table 1 plants-14-01829-t001:** Genomic identification and protein properties of *CsDOF* gene family in Tieguanyin.

Gene Accession	Gene ID	Size/aa	MW /Da	Theoretical pI	Instability Index	Aliphatic Index	GRAVY
CsTGY15G0001562a	*CsDOF*1	298	32,584.53	6.58	52.30	47.55	−0.804
CsTGY15G0001562b	*CsDOF*2	292	32,094.00	6.58	50.88	48.84	−0.824
CsTGY02G0001308a	*CsDOF*3	322	35,039.34	7.71	48.58	52.76	−0.754
CsTGY15G0000298b	*CsDOF*4	225	24,360.63	8.61	50.88	44.71	−0.829
CsTGY02G0002212b	*CsDOF*5	184	20,429.35	8.28	55.52	32.39	−1.091
CsTGY13G0002427a	*CsDOF*6	333	35,723.59	9.21	62.62	66.79	−0.602
CsTGY04G0002458a	*CsDOF*7	325	34,441.49	9.20	53.28	58.18	−0.517
CsTGY11G0000951b	*CsDOF*8	316	33,927.58	9.15	62.64	50.32	−0.672
CsTGY01G0000579a	*CsDOF*9	349	36,960.20	9.24	55.70	57.56	−0.516
CsTGY04G0003389a	*CsDOF*10	345	36,654.78	8.93	49.35	62.43	−0.488
CsTGY04G0003066a	*CsDOF*11	342	36,222.31	9.20	52.63	58.16	−0.554
CsTGY10G0001612b	*CsDOF*12	341	35,505.43	9.18	59.77	50.67	−0.577
CsTGY10G0001612a	*CsDOF*13	343	35,712.61	9.18	59.70	50.93	−0.573
CsTGY01G0002022b	*CsDOF*14	340	36,828.13	8.72	52.56	57.41	−0.551
CsTGY08G0000732a	*CsDOF*15	328	36,331.21	8.62	51.05	56.13	−0.772
CsTGY08G0000732b	*CsDOF*16	329	36,432.32	8.62	52.10	55.96	−0.772
CsTGY11G0000819b	*CsDOF*17	374	40,329.96	7.25	45.15	61.52	−0.533
CsTGY11G0000819a	*CsDOF*18	306	33,187.77	6.78	46.07	50.95	−0.720
CsTGY04G0002605b	*CsDOF*19	275	30,933.40	7.29	47.37	48.55	−0.878
CsTGY15G0001184a	*CsDOF*20	321	34,988.04	6.48	45.51	49.16	−0.668
CsTGY15G0001184b	*CsDOF*21	321	34,961.01	6.48	47.95	49.16	−0.660
CsTGY06G0002442a	*CsDOF*22	307	34,410.47	6.10	48.78	49.87	−0.738
CsTGY04G0001690a	*CsDOF*23	309	34,596.60	6.14	44.27	50.45	−0.768
CsTGY04G0001690b	*CsDOF*24	309	34,582.57	6.14	43.56	50.13	−0.769
CsTGY14G0001635b	*CsDOF*25	294	32,015.60	8.11	55.97	60.31	−0.594
CsTGY14G0001635a	*CsDOF*26	290	31,758.36	7.56	55.85	61.14	−0.596
CsTGY12G0002088a	*CsDOF*27	268	29,609.09	8.38	62.48	55.60	−0.708
CsTGY12G0002088b	*CsDOF*28	268	29,635.17	8.38	61.04	57.05	−0.691
CsTGY07G0002456a	*CsDOF*29	260	28,623.71	9.19	58.67	44.69	−0.855
CsTGY06G0001769a	*CsDOF*30	281	30,751.20	8.98	44.05	55.84	−0.686
CsTGY14G0002493a	*CsDOF*31	393	44,607.07	9.15	39.90	79.87	−0.338
CsTGY04G0001110b	*CsDOF*32	247	27,173.19	4.90	53.90	63.93	−0.526
CsTGY04G0001110a	*CsDOF*33	247	27,120.14	4.81	52.50	63.93	−0.498
CsTGY09G0001901b	*CsDOF*34	247	27,871.06	4.84	54.69	61.94	−0.644
CsTGY09G0001901a	*CsDOF*35	247	27,872.04	4.77	54.19	61.94	−0.644
CsTGY08G0001892b	*CsDOF*36	309	34,364.48	9.26	73.70	37.31	−1.201
CsTGY08G0001892a	*CsDOF*37	310	34,482.58	9.15	72.87	37.19	−1.204
CsTGY08G0000731a	*CsDOF*38	255	28,315.20	9.55	58.67	44.00	−1.046
CsTGY08G0000731b	*CsDOF*39	254	28,121.08	9.61	59.89	47.64	−1.009
CsTGY08G0000666b	*CsDOF*40	254	27,562.42	8.13	60.72	45.35	−0.78
CsTGY08G0000666a	*CsDOF*41	256	27,794.64	8.18	58.71	45.39	−0.784
CsTGY08G0001840b	*CsDOF*42	290	32,016.35	6.49	53.40	54.83	−0.678
CsTGY08G0001840a	*CsDOF*43	290	32,085.46	6.75	53.38	55.17	−0.683
CsTGY08G0002044b	*CsDOF*44	178	19,965.31	9.11	57.00	47.13	−0.803
CsTGY08G0000894b	*CsDOF*45	161	18,141.50	8.94	43.88	49.69	−0.684
CsTGY08G0000886b	*CsDOF*46	161	18,141.50	8.94	43.88	49.69	−0.684
CsTGY13G0000664a	*CsDOF*47	479	52,129.17	5.98	45.90	55.11	−0.653
CsTGY12G0000884b	*CsDOF*48	463	51,378.19	5.74	61.36	56.24	−0.799
CsTGY12G0000884a	*CsDOF*49	463	51,378.19	5.74	61.36	56.24	−0.799
CsTGY06G0000459a	*CsDOF*50	468	51,745.36	6.91	50.26	57.93	−0.825
CsTGY01G0003400a	*CsDOF*51	463	50,075.45	5.97	57.22	51.62	−0.846
CsTGY15G0001391a	*CsDOF*52	464	50,800.58	5.52	51.39	54.91	−0.844
CsTGY02G0001411b	*CsDOF*53	442	48,480.56	6.00	49.29	61.61	−0.721
CsTGY02G0001411a	*CsDOF*54	442	48,310.31	6.00	48.40	61.38	−0.704
CsTGY13G0002412b	*CsDOF*55	233	24,293.06	7.67	44.78	57.00	−0.358
CsTGY04G0000391a	*CsDOF*56	224	23,584.15	7.58	52.45	51.29	−0.521
CsTGY09G0002539b	*CsDOF*57	239	25,202.69	6.51	62.78	47.74	−0.654
CsTGY09G0002539a	*CsDOF*58	255	26,815.42	5.75	60.72	47.41	−0.652

## Data Availability

Data available in a publicly accessible repository. The original data presented in the study are openly available in the National Center for Biotechnology Information under accession number JAFLEL000000000 and in the GWH (https://bigd.big.ac.cn/gwh/, accessed on 3 April 2024) under the accession num-bers GWHASIV00000000.

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
