# Peer review of "Genome-Wide Characterization and Functional Analysis of CsDOF Transcription Factors in Camellia sinensis cv. Tieguanyin Under Combined Heat–Drought Stress"

_plants, 2025, doi:10.3390/plants14121829_

Round 1
Reviewer 1 Report
Comments and Suggestions for Authors
The structure of the manuscript plants-3651751 is unclear. When discussing the information in the subsections of the Results section, the results described in the previous subsections are almost never used, despite frequent semantic intersections, which create an effect of repetition rather than development of the discussion. At the same time, the order of the subsections should be changed in accordance with their logical connections with each other.
It is not clear from the manuscript what principle the authors used to number the genes and subfamilies of the Dof gene family. The numbering of Dof genes by genome, in my opinion, is chaotic, which complicates finding the localization of each gene. There is no information on localization in Table 1. As a result, in order to find the location of a gene on chromosomes, I need to study Figure 2 for a long time and carefully, the quality of which leaves much to be desired.
The authors should also have given an introductory description of the Camellia sinensis genome as a whole and the genome with which the work was carried out. Why was the genome from GenBank not used? In GenBank, the genome contains 15 chromosomes (ASM1731120v1), without the HA and HB variants. The manuscript does not explain why, out of 15 chromosomes of Camellia sinensis, the authors list 23 chromosomes in Figure 2, while in the text (line 164) 21 chromosomes are indicated.
The Discussion section is presented in an unacceptable form; the authors list their results (repeat of the Results section) without any correlation with the results of other researchers. And the literature data in the Discussion section are weakly related to the results presented in the manuscript. The Discussion section should be completely changed.
When reviewing the results, there are quite a few errors and controversial statements. For example, in line 123, “the B2 subfamily comprised only 1 member”, but in Figure 2, three genes are assigned to the B2 subfamily: CsDOF1, CsDOF2, and CsDOF26. Or, (lines 124-127 and further) the authors predict the function of genes based on homology with the described genes of other plants, but the involvement of transcriptional regulators in metabolism is largely determined not by their amino acid sequences but by the nucleotide sequences of their promoters, information about which the authors do not provide, although in subsection 4.6. (lines 605-610) they indicate that promoter analysis was carried out.
The manuscript also contains incorrect citations. For example, (line 357) why is a reference to the work of Rathnasamy et al., 2023, used here? This reference does not contain any information about the CsDOF5 gene.
The presentation and discussion of the results should be improved. So, how strong, in the authors' opinion, should the changes in metabolism be with a change in gene activity of less than 2 times (Figure 8)? Or, the authors in Figure 3 discuss domains in the protein products of genes but do not indicate them in the 3D models of these same proteins in Figure 7. I did not see in Figure 7 the domains common to all CsDOF proteins – additional indication of these structures is necessary.
Reviewer 2 Report
Comments and Suggestions for Authors
I have reviewed this manuscript carefully. The authors identified the transcription factor DOF family genes in TieGuanyin Tea and used QRT-PCR analysis to evaluate the expression profile of 13 CsDOF genes under high temperature and drought conditions. There are many questions to be answered by the author. My main comments are as follows:
Major concerns:
- What is the basis for subfamily classification? I haven't seen a specific explanation?
- In line 135, Chz04HA and Chz08HA, why didn't I see these two chromosomes? Did the author make a mistake? It should be carefully checked.
- Figure 3 is not clear. The characters are too small and the figure is not standardized.
- In line 197, The author analyzed the expression of the DOF gene in roots, stems and leaves, but why only these three tissues were selected? The author should specify in detail in the method what stages the materials of the roots, stems and leaves are respectively.
- In Figure 5, the tandem and segmental duplication'sshould be represented by lines of different colors.
- In line 245-270, the author needs to analyze the specific amount of collinearity among different species and provide specific data. It is too general to just say that it has increased or decreased.
- The result in Figure 7 is not very consistent with the results in the previous and subsequent texts. It is suggested to attach it and increase the clarity of the figure.
- In line 305, did the author analyze the expression changes of the DOF gene under light stress treatment? Why haven't I seen the specific data or the graph? Also, in the method, conditions such as drought, high temperature, and light stress should be clearly stated, along with the specific size of the plants and how they were treated.
Minor concerns:
- In line 39, there is an extra period. It should be deleted.
- In line 172, CSDOF should be CsDOF.
- The reference formats are inconsistent. For instance, 25 has no page numbers, and there are many other inconsistent formats before and after. It is hoped that the author will carefully check and maintain one format.
The language needs further polishing and processing.
Round 2
Reviewer 1 Report
Comments and Suggestions for Authors
The authors have made changes in accordance with my comments, but I find these changes insufficient. In particular,
The description of the Camellia sinensis cv. Tieguanyin genome in the manuscript does not explain what the authors mean by the chromosome designations “HA” and “HB”. Why are the chromosomes designated as “Chz” on line 116 and as “Chr” elsewhere? The information on the genomes used (section 4.1) does not contain direct references to nucleotide sequences or project descriptions, which makes it impossible to use the manuscript in further research.
The gene accessions do not allow for analysis of the nucleotide sequences, which is why the reliability of the results cannot be assessed.
The revised manuscript still does not explain why the DOF genes on chromosome 1 are designated as CsDOF9 and CsDOF51, while the authors designated the gene located on chromosome 15 as CsDOF1.
Section 2.2: In line 169, the authors indicate that the division of the DOF genes into 9 subfamilies was carried out in accordance with previously published data, but in the Results section, or in the Discussion section, these previously published data are not presented or discussed. An analysis of Figure 2 gives reason to doubt the correctness of the assignment of genes to subfamilies. On what basis are the CsDOF40-CsDOF43 genes assigned to subfamily D1? In Figure 2, the branch for these 4 genes separates from the root, which does not allow these genes to be assigned to the same subfamily with other CsDOF genes.
The revised Discussion section still does not compare the authors' results with previously published data.
In addition, the manuscript contains terminology errors. For example, (line 123) the authors confuse the concepts of "gene" and "gene product"; Figure 3: for the first and second columns, the scale is given for amino acids, but the designations 5' and 3' are for nucleic acids; (line 375) glutamine is an amino acid, not an enzyme.
Reviewer 2 Report
Comments and Suggestions for Authors
The authors have accordingly revised the manuscript. I have no further comments.
Author Response
Thank you for your review and approval.
Round 3
Reviewer 1 Report
Comments and Suggestions for Authors
In section 2.2, add a discussion of the uniqueness of the subfamily D1.1 genes of Camellia sinensis.
Section 4.1 should provide the full scientific names of the plant species: Camellia sinensis Kuntze, Oryza sativa L., Solanum tuberosum L., Zea mays L., Arabidopsis thaliana (L.) Heynh.
